# Husk of Agarwood Fruit-Based Hydrogel Beads for Adsorption of Cationic and Anionic Dyes in Aqueous Solutions

**DOI:** 10.3390/molecules26051437

**Published:** 2021-03-06

**Authors:** Chih Ming Ma, Bo-Yuan Yang, Gui-Bing Hong

**Affiliations:** 1Department of Cosmetic Application and Management, St. Mary’s Junior College of Medicine, Nursing and Management, Yilan 266, Taiwan; cmma@smc.edu.tw; 2Department of Chemical Engineering and Biotechnology, National Taipei University of Technology, Taipei 106, Taiwan; toro520191@gmail.com

**Keywords:** agarwood, hydrogel, adsorption, reactive blue 4, crystal violet

## Abstract

Hydrogel beads based on the husk of agarwood fruit (HAF)/sodium alginate (SA), and based on the HAF/chitosan (CS) were developed for the removal of the dyes, crystal violet (CV) and reactive blue 4 (RB4), in aqueous solutions, respectively. The effects of the initial pH (2–10) of the dye solution, the adsorbent dosage (0.5–3.5 g/L), and contact time (0–540 min) were investigated in a batch system. The dynamic adsorption behavior of CV and RB4 can be represented well by the pseudo-second-order model and pseudo-first-order model, respectively. In addition, the adsorption isotherm data can be explained by the Langmuir isotherm model. Both hydrogel beads have acceptable adsorption selectivity and reusability for the study of selective adsorption and regeneration. Based on the effectiveness, selectivity, and reusability of these hydrogel beads, they can be treated as potential adsorbents for the removal of dyes in aqueous solutions.

## 1. Introduction

With the rapid development of industry and population growth, natural resources cannot meet the demand of human life. The extensive use of synthetic materials and the discharge of wastewater from manufacturing industries have seriously polluted water resources and the environment. Industrial wastewater contains organic compounds and heavy metals, which are water pollutants that cause worldwide environmental problems [1]. Synthetic dyes are common organic pollutants in industrial wastewater. Most of these dyes have diverse structures and stable properties and are not easily biodegraded [2]. The effective removal of dyes from wastewater is one of the major environmental challenges in water management since many dyes are carcinogenic, mutagenic, allergenic, and highly toxic to aquatic living organisms [3]. The methods developed for treating pollutants in wastewater currently include adsorption, membrane separation, reverse osmosis, photodegradation, ozone treatment, oxidation treatment, biological treatment, coagulation and flocculation, chemical conversion, electrochemical methods, etc. [4]. The adsorption method for wastewater treatment has the advantages of inexpensiveness, high efficiency, easy operating procedures, system flexibility, wide-range applicability, insensitivity to pollutants, and facilitation of material recovery [5,6]. Various low-cost, environmentally friendly alternative adsorbents have been prepared from agricultural waste or byproducts for the removal of dyes from wastewater, such as mung bean husk [7], corn stalk [8], rice bran [9], bamboo [10], rice husk [11], and Artocarpus odoratissimus leaves [12]. These green adsorbents have the major characteristic properties of low cost with satisfactory adsorption capacities and environmentally friendly nature (such as derived from abundant natural sources, biodegradable, non-toxic, etc.), which have the lowest impact on the environmental ecosystem [13].

Recently, hydrogel-based adsorbents synthesized from bioresources such as chitosan, sodium alginate, and starch cellulosic materials have attracted particular attention for dye removal because of their three-dimensional network structure, high availability, environmental friendliness, high swelling property, versatility as a treatment medium, and simple recoverability [14,15,16]. Sodium alginate (SA) is a natural polysaccharide composed of β-d-mannuronate units and α-l-guluronate units and can be extracted from seaweed [17]. Chitosan (CS), a copolymer polysaccharide of deacetylated units and acetylated units, is prepared mainly by deacetylation of chitin [18]. SA has a high content of carboxyl groups, and CS has a large number of amino (-NH_2_) and hydroxyl (-OH) groups that can be chemically or physically reacted with a crosslinking agent to prepare hydrogels [19,20]. In recent years, many studies have pointed out that hydrogels based on SA and CS have good adsorption capacities for pollutants such as metal ions, antibiotics, and organic pollutants in aqueous media and are efficient for wastewater treatment [21,22,23,24]. However, the low mechanical strength of hydrogels limits its commercial applications as adsorbents. Agarwood is the most expensive wood in the world and is widely planted in Southeast Asia [25]. Agarwood fruit has a pear-like shape and a hollow structure inside. The husk of agarwood fruit (HAF) is quite hard and generally discarded as agricultural waste that is used due to good mechanical strength, superior adsorption potential, and availability [26]. The introduction of HAF into chitosan may improve the swelling strength and stability of the adsorbent. Crystal violet (CV) dye is a tri-phenylmethane dye and reactive blue 4 (RB4) dye is an anthraquinone-based chlorotriazine dye. Both of them are often used in the textile industry and are known for their carcinogenic, mutagenic, and mitotic poisoning nature [27,28]. Therefore, this work attempts to prepare hydrogel beads based on the husk of agarwood fruit, SA, and CS for the removal of CV and RB4 dyes in aqueous solutions. In addition, the kinetics, isotherms, selectivity, and regeneration ability of the hydrogel beads were investigated to evaluate their adsorption properties.

## 2. Results and Discussion

### 2.1. Characterization

Scanning electron microscopy (SEM) was applied to study the morphologies of the hydrogel beads. Figure 1 shows the SEM images of hydrogel beads. It can be seen that the particle size of all the hydrogel beads is approximately 1.5 mm. Before chemical modification, the surface of HAF-SA is smooth, and that of HAF-CS is rough (Figure 2a,b). After modification, the surface of HAF-SA becomes wrinkled, and that of HAF-CS becomes smooth (Figure 2c,d). In addition, more clearly spherical shapes could be observed for the MHAF-CS hydrogel beads. From the high-magnification SEM images, the wrinkled surface of hydrogel beads was observed due to the incorporation of HAF into the hydrogel beads. The functional groups in HAF interacted strongly with SA and CS and resulted in a higher crosslinking degree [29].

FTIR analysis was conducted to identify the functional groups on the surface of the hydrogel beads. The FTIR spectra of the MHAF-SA hydrogel beads before and after CV adsorption are shown in Figure 1a. In the spectrum, the broad peak of MHAF-SA at 3370 cm^−1^ was attributed to the stretching vibrations of the –OH groups from pure SA and HAF molecules. The peaks at 1610 cm^−1^, 1410, and 1028 cm^−1^ are assigned to the –COO^−^ (asymmetric), –COO^−^ (symmetric), and C-O-C stretching vibrations, respectively [30]. After modification by acetic acid, the intensity of these peaks in the spectrum of the MHAF-SA hydrogel beads was increased compared with that of the HAF-SA hydrogel beads. In addition, new peaks were generated at 1718 cm^−1^ (C=O) and 1238 cm^−1^ (C-O), indicating more incorporation of –COO^−^ groups. All the peaks associated with –COO^−^ shifted in the spectra of MHAF-SA after CV adsorption and indicated electrostatic interactions between the negatively charged carboxyl groups of MHAF-SA and the positively charged CV dye. Figure 1b displays the FTIR spectra of MHAF-CS hydrogel beads before and after RB4 adsorption. The broad peak of MHAF-CS at approximately 3350 cm^−1^ can be assigned to the overlapping OH and NH stretching vibrations contributed by pure CS and HAF. Other characteristic peaks at 2872, 1590, 1373, and 1027 cm^−1^ are attributed to C-H stretching, N-H bending, C-H bending, and C-O-C stretching vibrations, respectively. After modification by sodium hydroxide, the peaks at 3350, 1590, and 1027 cm^−1^ were obviously enhanced compared with those of HAF-CS. After RB4 adsorption, the peaks at 3350, 1590, and 1027 cm^−1^ shifted to 3372, 1567, and 1016 cm^−1^, respectively. In addition, a new peak appears at 1160 cm^−1^ after RB4 adsorption, which corresponds to the −SO_3_^−^ groups of RB4 molecules [31]. According to this information, RB4 adsorption onto MHAF-CS hydrogel beads may be dependent on –OH and –NH_2_ groups.

Figure 3 shows the point of zero charge (pH_pzc_) of the MHAF-SA and MHAF-CS hydrogel beads. The pH_pzc_ values of the MHAF-SA and MHAF-CS hydrogel beads were 2.50 and 7.35, respectively. Therefore, the surface of MHAF-CS was more positive than MHAF-SA due to the presence of –NH_2_ groups, as proved by the FTIR spectra. MHAF-SA can be used as a cation attractor in solutions with pH values above the pH_pzc_, and MHAF-CS can be treated as an anion attractor in solutions with pH values below the pH_pzc_ [32]. The swelling behavior of hydrogel beads was studied, and the results are shown in Figure 4. As displayed in Figure 4, compared to CS, sodium alginate (SA) exhibits a much lower swelling ratio (*S*%) and does not change much with increasing time. The surface of the SA hydrogel bead is hydrophobic, and water molecules do not easily enter the interior. Thus, the swelling ratio of SA is low and does not change obviously (Figure 4a). The swelling ratio of MHAF-SA increased significantly with the addition of HAF due to certain hydrophilic groups in HAF allowing water molecules to enter the beads. The higher swelling property can improve dye adsorption since more water molecules can penetrate into the beads [33,34]. The chitosan structure can generate hydrogen bonds with water molecules. Therefore, the chitosan (CS) hydrogel beads have good water adsorption capacity. The swelling ratio of MHAF-CS decreased with the addition of HAF, which is less hydrophilic than chitosan (Figure 4b). Although the swelling ratio of MHAF-CS is lower than that of chitosan, it was better for adsorption performance since the introduction of HAF into chitosan improved the swelling strength and stability [34].

### 2.2. Effect of pH

The pH of the solution is one of the most important influencing factors in the adsorption system and can change the surface charge of the adsorbent. The pH effect on the dye adsorption of hydrogel beads at a fixed dosage of 30 mg and an initial dye concentration of 200 mg/L is shown in Figure 5. The adsorption capacity of CV adsorbed onto MHAF-SA increased as the pH increased, while RB4 adsorbed onto MHAF-CS had the opposite effect. At pH values higher than 2.5 (pH_pzc_ of MHAF-SA), the surface of the hydrogel bead is negatively charged and prefers to adsorb the cationic dye (CV). The surface of the adsorbent is occupied by a large amount of H^+^ ions under acidic conditions, which compete for the adsorption sites with CV and result in a lower adsorption capacity. Moreover, the cationic dye easily reacts with oxygenic functional groups (–COO^−^ and –OH^−^ groups) in a high pH dye solution [30]. At pH values lower than 7.35 (pH_pzc_ of MHAF-CS), the surface of the hydrogel bead is positively charged due to the protonation of amino groups (–NH_3_^+^) and facilitates RB4 (SO_3_^−^ groups) adsorption by electrostatic attraction. MHAF-CS tends to dissolve at pH 2, which reduces the adsorption capacity of RB4. In addition, the precipitation of CV was observed at pH 10. The optimum pH values for the adsorption of CV and RB4 were determined to be 9 and 3 for MHAF-SA and MHAF-CS, respectively.

### 2.3. Effect of Dosage

The dosage of the adsorbent is another important key factor that affects the adsorption system to reduce the cost of wastewater treatment and maximize the benefit of the adsorbent. As shown in Figure 6, the removal efficiency (*RE* %) of both CV and RB4 increased with the increasing dosage, while the adsorption capacity was the opposite. When the dosage of hydrogel beads was 2.5 g/L, the removal efficiencies of CV and RB4 increased insignificantly with increasing dose. Therefore, the optimum dosage for both hydrogel beads to adsorb dyes was 2.5 g/L.

### 2.4. Effect of Contact Time

The effect of contact time on the adsorption of CV and RB4 onto hydrogel beads was investigated with a dosage of 2.5 g/L and an initial dye concentration of 200 mg/L at optimum pH values and 40 °C. Figure 7 shows the influence of the contact time on the dye adsorption capacities of MHAF-SA and MHAF-CS. Both hydrogel beads have similar trends for adsorption capacity, which increases with time. The adsorption, in the beginning, was very fast since a large number of available active sites or free functional groups can attract dye molecules, resulting in the adsorption capacities increasing significantly. Subsequently, the adsorption of CV and RB4 onto MHAF-SA and MHAF-CS became slower and tended to be stable, respectively, after 360 min and 420 min, finally approaching an equilibrium state. The slow adsorption can be ascribed to the active sites or free functional groups gradually being filled and the accumulation of dye molecules in the hydrogel beads, which makes dye adsorption onto the hydrogel beads more difficult.

### 2.5. Adsorption Kinetics

To study the effectiveness of the hydrogel beads, the adsorption kinetics of CV and RB4 onto the hydrogel beads were analyzed with pseudo-first-order model (PFOM, expressed by Equation (1)) and pseudo-second-order model (PSOM, expressed by Equation (2)) [35,36].
(1)$$\ln\left(q_{e}-q_{t}\right)=\ln q_{e}-k_{1}t$$
(2)$$\frac{t}{q_{t}}=\frac{1}{k_{2}q_{e}^{2}}+\frac{t}{q_{e}}$$ where *k*_1_ (1/min) and *k*_2_ (g/mg·min) are the adsorption rate constants of pseudo-first-order and pseudo-second-order models, respectively; *q**_t_* (mg/g) and *q**_e_* (mg/g) are the adsorption capacities at time *t* (min) and adsorption equilibrium, respectively. 

The parameters of the kinetic models and correlation coefficients are summarized in Table 1. As shown in Figure 8, the pseudo-second-order model is more suitable than the pseudo-first-order model to describe the kinetic data of CV onto MHAF-SA hydrogel beads. However, compared to the pseudo-second-order model, the pseudo-first-order model yields a better fit for RB4 onto MHAF-CS. The results indicated that the kinetic data of CV adsorption onto MHAF-SA agreed well with the calculated values by the pseudo-second-order model, and the adsorption behavior for RB4 onto MHAF-CS obeyed the pseudo-first-order model. The pseudo-first-order model describes the adsorption in solid-liquid systems based on the sorption capacity of solids and the pseudo-second-order kinetics model represents chemisorption that involved chemical reactions between adsorbent and adsorbate [37].

### 2.6. Adsorption Isotherms

The adsorption isotherm study provides details about the interaction between the dye molecules and hydrogel beads at equilibrium. Various initial CV and RB4 dye concentrations (20–600 mg/L) at constant temperature were evaluated. The linear forms of isotherm models such as the Langmuir (expressed by Equation (3)) [38] and Freundlich (expressed by Equation (4)) [39] models were used to fit the adsorption isotherm data.
(3)$$\frac{C_{e}}{q_{e}}=\frac{1}{q_{m}K_{L}}+\frac{C_{e}}{q_{m}}$$
(4)$$\ln q_{e}=\frac{1}{n}\ln C_{e}+\ln K_{F}$$
where *C_e_* (mg/L) is the dye equilibrium concentration in aqueous solution; *q_m_* (mg/g) and *q_e_* (mg/g) are the maximum adsorption capacity and adsorption equilibrium, respectively; *K_L_* (L/mg) is the Langmuir constant; *K_F_* and *n* are the Freundlich isotherm constants and adsorption intensity, respectively. The plots are illustrated in Figure 9, and the corresponding isotherm parameters are shown in Table 2. 

Based on the linear regression coefficient (*R*^2^), the adsorption of CV onto MHAF-SA and RB4 onto MHAF-CS correlated well with the Langmuir isotherm model, indicating that the dyes CV and RB4 were mainly adsorbed onto a homogeneous surface of hydrogel beads and had monolayer coverage with uniform adsorption energies. The maximum monolayer adsorption capacity (*q_m_*) values from the Langmuir isotherm model for CV adsorbed onto MHAF-SA and RB4 adsorbed onto MHAF-CS were compared with other adsorbents reported in the literature and presented in Table 3. As can be seen from this table, the MHAF-SA and MHAF-CS showed higher adsorption capacity for CV and RB4, suggesting that these adsorbents can be applied potentially in the adsorption system.

### 2.7. Selective Adsorption of Hydrogel Beads

In real applications, the selective removal of pollutants from wastewater is important [32]. Three groups of dye mixtures, namely, RB4/CR, CV/NR, and RB4/CV, were used to study the selectivity of the MHAF-SA and MHAF-CS hydrogel beads. As shown in Figure 10a, the absorption peaks of the mixed dyes (RB4/CR) showed no obvious change after adsorption. In addition, the peak shape was not similar to any absorption spectra of a single dye, which means that MHAF-SA has poor adsorption capacity for anionic dyes due to the electrostatic repulsion between the dyes and hydrogel beads. Figure 10b shows the absorption spectra results of MHAF-SA before and after adsorption of the cationic mixed dyes (CV/NR). After adsorption, the intensity of the absorption peaks of the mixed dyes decreased obviously, indicating that MHAF-SA has good adsorption performance for cationic dyes. As presented in Figure 10c, the absorption peak corresponding to CV (cationic dye) in the mixed dye solution disappeared after adsorption, demonstrating that the cationic dye could be selectively removed by MHAF-SA. Figure 10d shows the results of RB4/CR adsorbed onto MHAF-CS. After adsorption, the intensity of the mixed dyes decreased, and the absorption spectrum was not similar to that of any single dye, indicating that MHAF- CS can effectively adsorb anionic dyes containing SO_3_^−^ groups. As shown in Figure 10e, there was no obvious change in the mixed dye (CV/NR) before and after adsorption, implying that MHAF-CS does not support the adsorption of cationic dyes. Figure 10f shows the change in the absorption peak before and after the adsorption of the mixed dye (RB4/CV). After absorption, the absorption peak intensity of the mixed dye decreased, and the absorption spectra were close to those of the CV dye. Therefore, MHAF-SA and MHAF-CS have acceptable adsorption selectivity towards cationic and anionic dyes, respectively.

### 2.8. Regeneration of Hydrogel Beads

The regeneration performance of the hydrogel beads is shown in Figure 11. As presented in this figure, the CV removal efficiency of MHAF-SA is not obviously decreased for up to five cycles, indicating that MHAF-SA has high reusability. Although the RB4 removal efficiency of MHAF-CS decreased by approximately 20% under the same conditions, MHAF-CS still retained an *RE* (%) over 70%, suggesting the reusability of MHAF-CS hydrogel beads. The reusability of MHAF-SA and MHAF-CS is also compared with the report in the literature, as shown in Table 3. The reusability of MHAF-SA is excellent for CV removal, and there is acceptable reusability of MHAF-CS for RB4 removal. These results indicate that the hydrogel beads prepared in this work can be regenerated and reused to efficiently and selectively remove CV and RB4 dyes.

### 2.9. Adsorption Mechanism

According to the FTIR spectra of MHAF-SA and MHAF-CS before and after adsorption of dyes, the effect of pH discussion, and based on the surface properties of the adsorbents, the adsorption mechanism of CV onto MHAF-SA and RB4 onto MHAF-CS are illustrated in Figure 12. The CV adsorption onto the surface of MHAF-SA may be due to the cationic dye easily reacting with oxygenic functional groups (–COO^−^ and –OH^−^ groups) in a high pH dye solution (see Figure 12a). Under acidic conditions, the surface of MHAF-CS is positively charged due to the protonation of amino groups (–NH^3+^) and facilitates RB4 (SO_3_^−^ groups) adsorption by electrostatic attraction (Figure 12b). Therefore, the adsorption of CV onto MHAF-SA and RB4 onto MHAF-CS might contain the electrostatic attraction and hydrogen bonding combination [40,49].

## 3. Materials and Methods

### 3.1. Materials

Acetic acid (purity ≧ 99.83%), ethanol (purity ≧ 99.8%), ammonia hydroxide (purity ≧ 35%), and sodium hydroxide (purity ≧ 97%) were purchased from Fisher Chemical (Pittsburgh, PA., USA). Sodium chloride (purity > 99.5%), Congo red (CR), and crystal violet (CV) were purchased from Acros Organics (Geel, Belgium). Alginic acid sodium salt powder, chitosan, reactive blue 4 (RB4), and neutral red (NR) were supplied by Sigma-Aldrich (Natick, MA., USA). Hydrochloric acid (purity > 37%) was obtained from Scharlau (Senmanat, Spain). Calcium chloride (purity ≧ 89%) was purchased from Avantor (Allentown, PA., USA). All chemicals were analytical grade and used without further purification. Agarwood (*Aquilaria agallocha* Roxb) fruit was obtained from the local farmer in Kaohsiung, Taiwan.

### 3.2. Agarwood Fruit Treatment

The treatment process of agarwood fruit is shown in Figure 13. First, the agarwood fruit was washed repeatedly with water four to five times to remove surface dirt. After cleaning, the fruit was exposed to the sun for 3–4 days to dry naturally. The husk was obtained from the fruit and was ground and sieved (<100 mesh) for adsorbent preparation.

### 3.3. Preparation of HAF-SA Hydrogel Beads

HAF powder (400 mg) was added to 40 mL of deionized water and stirred on a magnetic stirrer for 30 min. Then, the solution (10 mL) was mixed with 200 mg of sodium alginate powder and stirred continuously for 1 h to completely dissolve the powder. The solution was dropped into a calcium chloride solution (5 wt%) via a syringe needle at a speed of 75 mL/h. The HAF-SA hydrogel beads were formed by an ion exchange process. After washing with deionized water, the hydrogel beads were immersed in 4.4 M acetic acid solution at 30 °C for 5 h at an agitation rate of 100 rpm in a shaker with a thermostatic water bath. Then, the modified HAF-SA (MHAF-SA) hydrogel beads were filtered and washed with deionized water at least three times. Finally, the MHAF-SA hydrogel beads were obtained after drying in an oven at 50 °C for 6 h.

### 3.4. Preparation of HAF-CS Hydrogel Beads

The method to prepare chitosan hydrogels by physical crosslinking is preferred over the chemical crosslinking method due to the prevention of potential toxicity from chemical crosslinking agents [50]. Therefore, physically crosslinked chitosan hydrogels were applied in this study. First, chitosan solution was prepared by dissolving 1.2 g of chitosan powder in 50 mL of (1%, *v*/*v*) acetic acid solution. The solution was stirred continuously at 40 °C for 2 h and then remained undisturbed for 12 h to release air bubbles. Next, HAF powder (400 mg) was added to the chitosan solution and stirred for 30 min. The solution was dropped into a 0.5 M sodium hydroxide solution via a syringe needle with a speed of 75 mL/h. The hydrogel beads were formed and stirred for 3 h. Subsequently, the HAF-CS hydrogel beads were filtered, washed, and immersed in 4.4 M ammonia solution at 30 °C for 5 h at an agitation rate of 100 rpm in a shaker with a thermostatic water bath. After filtration, washing, and drying at 50 °C for 6 h, modified HAF-CS (MHAF-CS) hydrogel beads were obtained.

### 3.5. Characterization

The morphologies of the hydrogel beads were observed by scanning electron microscopy (SEM), and the functional groups on the surface were analyzed by Fourier transform infrared (FTIR) spectroscopy. The point of zero charge (pH_pzc_) was determined using the pH-drift method [51] to investigate the surface charge of the hydrogel beads. To test the swelling ability of the hydrogel beads, a gravimetric method was applied [52]. A certain amount of hydrogel beads was immersed in deionized water at room temperature for a period of 6 h. The swelling ratio (*S*%) was determined by the following equation [53]:
(5)$$S(\%)=\frac{W_{w}-W_{d}}{W_{d}}\times 100\%$$ where *W_w_* and *W_d_* are the weights (g) of wet and dried hydrogel beads, respectively.

### 3.6. Batch Adsorption Experiments

The dye adsorption experiments were performed on a BT-150D shaker (YIH DER Co., Ltd., Taiwan) with a shaking speed of 100 rpm via a batch adsorption process. Equilibrium adsorption experiments were carried out by adding 30 mg of hydrogel beads into 20 mL of different concentrations of dye solution at 30 °C. After the adsorption approached equilibrium, the content of dye adsorbed onto the hydrogel beads was analyzed at the maximum wavelength (RB4 = 595 nm; CV = 590 nm) by using a UV/VIS spectrophotometer (Thermo Scientific, Waltham, MA., USA). The pH effect was investigated by adjusting the initial pH of RB4 or CR solutions in the range of 2–10 while the initial dye concentration was fixed at 200 mg/L. The dosage effect of hydrogel beads was studied from 0.5 to 3.5 g/L with an initial dye concentration of 200 mg/L. The temperature effect on the adsorption was studied from 30 to 60 °C with an initial dye concentration of 200 mg/L. The isothermal adsorption experiments were conducted with 20 mL of dye solutions with different initial concentrations (200–600 mg/L) at a fixed temperature (30 °C). All adsorption experiments were carried out in triplicate to confirm the accuracy of this study. The adsorption capacity (*q*) and the removal efficiency (*RE* %) of RB4 or CR adsorbed onto hydrogel beads RB4 were calculated by the following equations:
(6)$$q=\frac{(C_{0}\times C_{e})V}{W}$$
(7)$$\mathrm{RE}(\%)=\frac{C_{0}\times C_{e}}{C_{0}}\times 100$$ where *C*_0_ and *C_e_* are the initial and equilibrium dye concentrations (mg/L), respectively, *V* (L) is the volume of the dye solution, and *W* (g) is the dry weight of the hydrogel beads used.

### 3.7. Selective Adsorption Study

Textile wastewater contains different types of dyes; thus, selective adsorption studies of adsorbents are quite important for practical applications. Three groups of dye mixtures, namely, RB4/CR (anionic dye-anionic dye), CV/NR (cationic dye-cationic dye), and RB4/CV (anionic dye-cationic dye), were used to study the adsorption selectivity of MHAF-SA and MHAF-CS hydrogel beads. The optimum dosage of hydrogel beads (2.5 g/L) was added to 20 mL of dye solution (200 mg/L) at a fixed temperature. After the adsorption reached equilibrium, the residual concentration of the dye mixture was monitored by a UV/VIS spectrophotometer.

### 3.8. Regeneration Study

To reduce the damage to the hydrogel beads during the regeneration process, the agents used in the preparation process were selected to perform dye desorption experiments. The dye-adsorbed MHAF-SA and MHAF-CS hydrogel beads were immersed in 4.4 M acetic acid solution and 4.4 M ammonia solution, respectively. After shaking for 10 min, the hydrogel beads were washed several times and then dried at 50 °C for 6 h to obtain the regenerated hydrogel beads.

## 4. Conclusions

Husk of agarwood fruit-based hydrogel beads (MHAF-SA and MHAF-CS) were successfully synthesized in this study. The functional groups of the hydrogel beads were identified and proved by the FTIR spectra. The dynamic adsorption behavior of CV onto MHAF-SA and RB4 onto MHAF-CS can be represented well by the pseudo-second-order model and pseudo-first-order model, respectively. The adsorption isotherm data of CV onto MHAF-SA and RB4 onto MHAF-CS can be explained by the Langmuir isotherm model, with *q_m_* values of 232.56–370.37 and 156.25–270.27 mg/g, respectively. Both hydrogel beads have excellent adsorption selectivity. After five adsorption-desorption cycles, the hydrogel beads still have a high removal efficiency. Overall, the hydrogel beads synthesized in this study have effectiveness, selectivity, and reusability and can be applied to the removal of dyes in aqueous solutions.

## Figures and Tables

**Figure 1 molecules-26-01437-f001:**
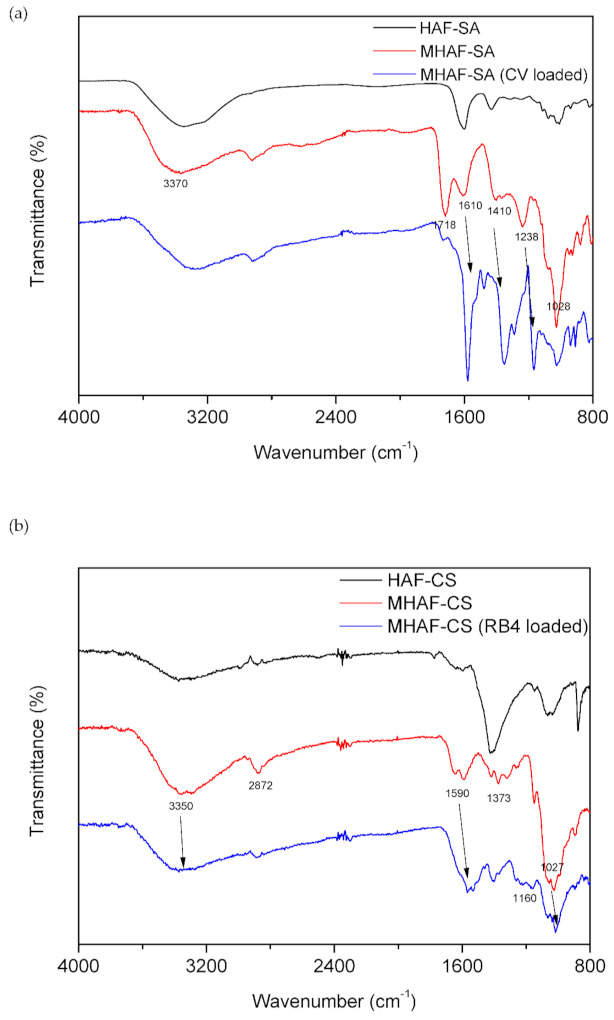
FTIR spectra of (**a**) MHAF-SA hydrogel beads and (**b**) MHAF-CS hydrogel beads.

**Figure 2 molecules-26-01437-f002:**
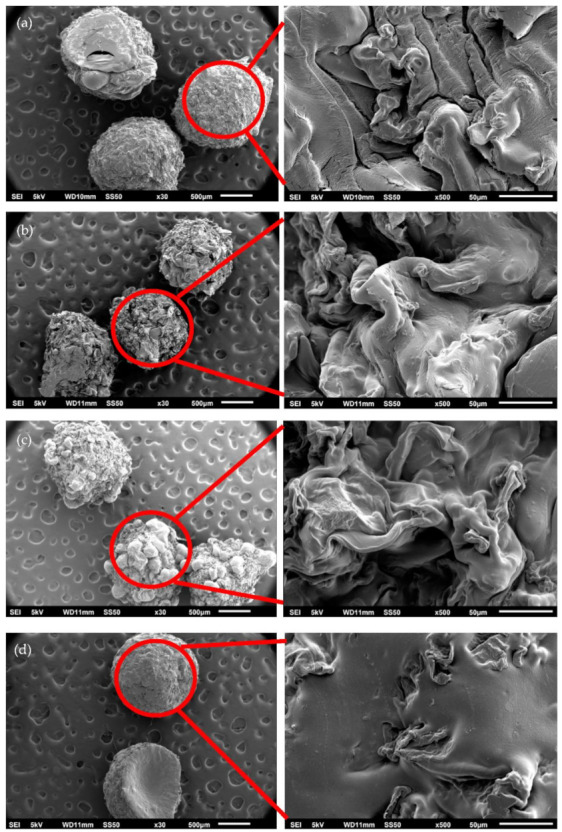
SEM images of hydrogel beads: (**a**) HAF-SA, (**b**) HAF-CS, (**c**) MHAF-SA, and (**d**) MHAF-CS.

**Figure 3 molecules-26-01437-f003:**
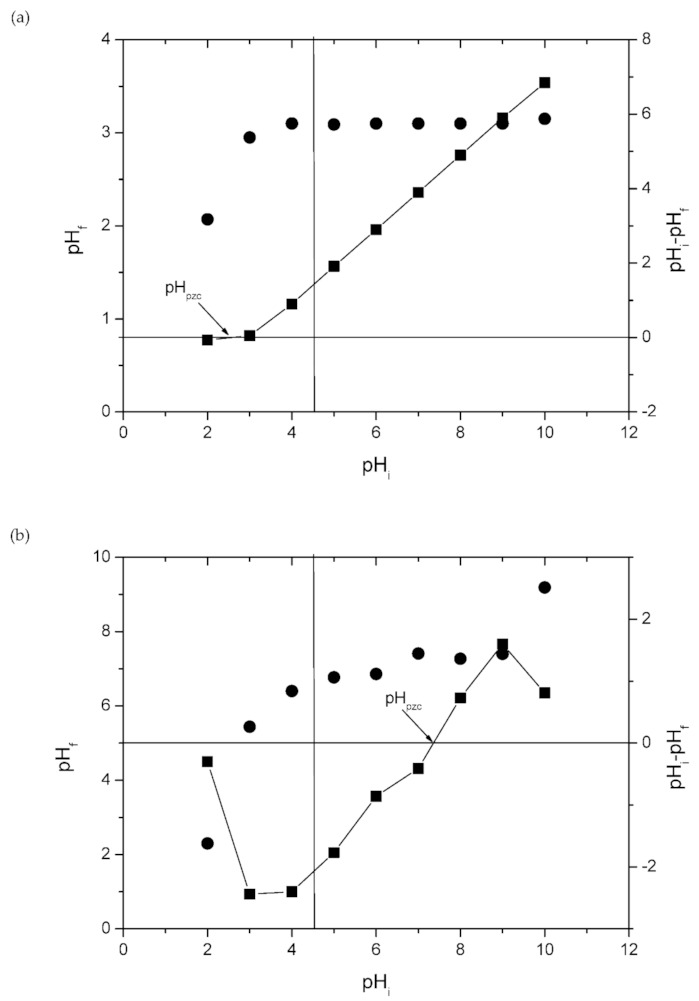
The pH_pzc_ of (**a**) MHAF-SA hydrogel beads and (**b**) MHAF-CS hydrogel beads.

**Figure 4 molecules-26-01437-f004:**
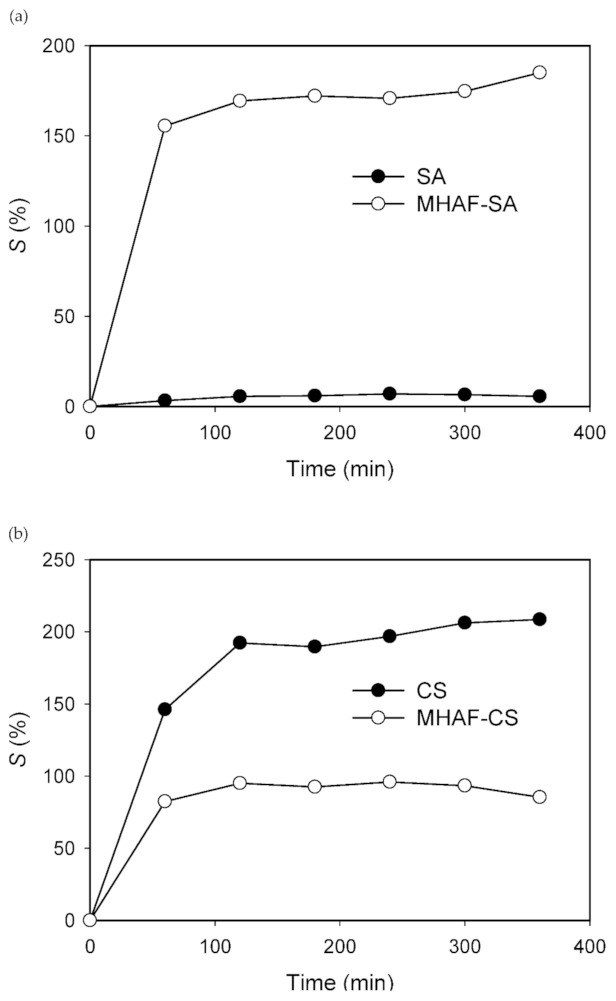
The swelling behaviors of (**a**) MHAF-SA hydrogel beads and (**b**) MHAF-CS hydrogel beads.

**Figure 5 molecules-26-01437-f005:**
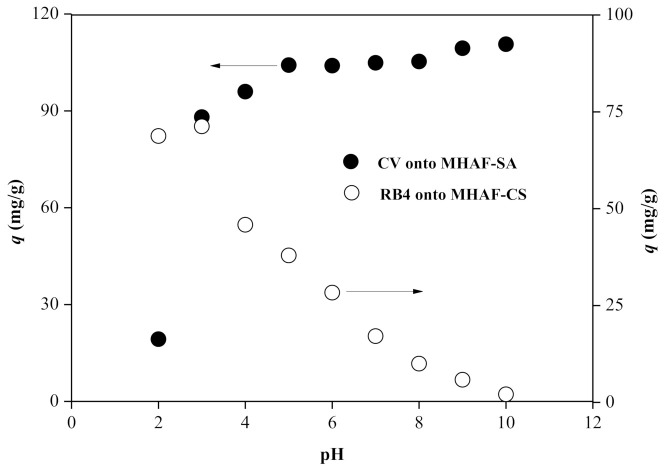
Effect of pH on dye adsorption.

**Figure 6 molecules-26-01437-f006:**
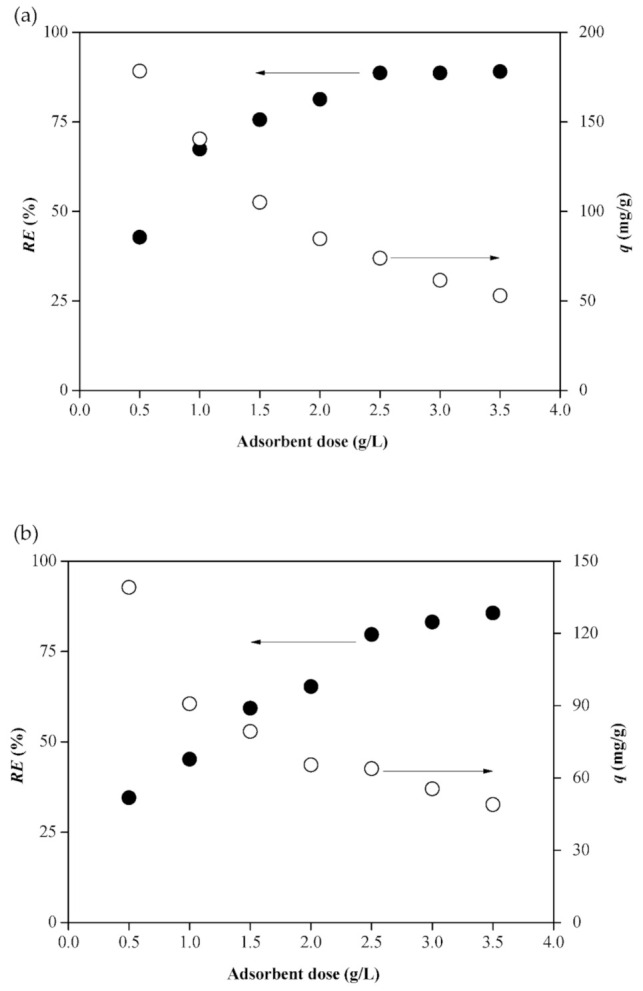
Effect of adsorbent dosage on the removal of (**a**) CV onto MHAF-SA and (**b**) RB4 onto MHAF-CS.

**Figure 7 molecules-26-01437-f007:**
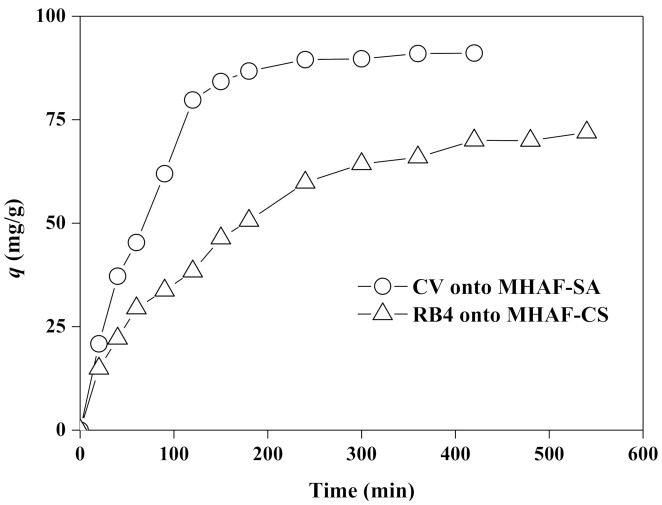
Effect of contact time on dye adsorption.

**Figure 8 molecules-26-01437-f008:**
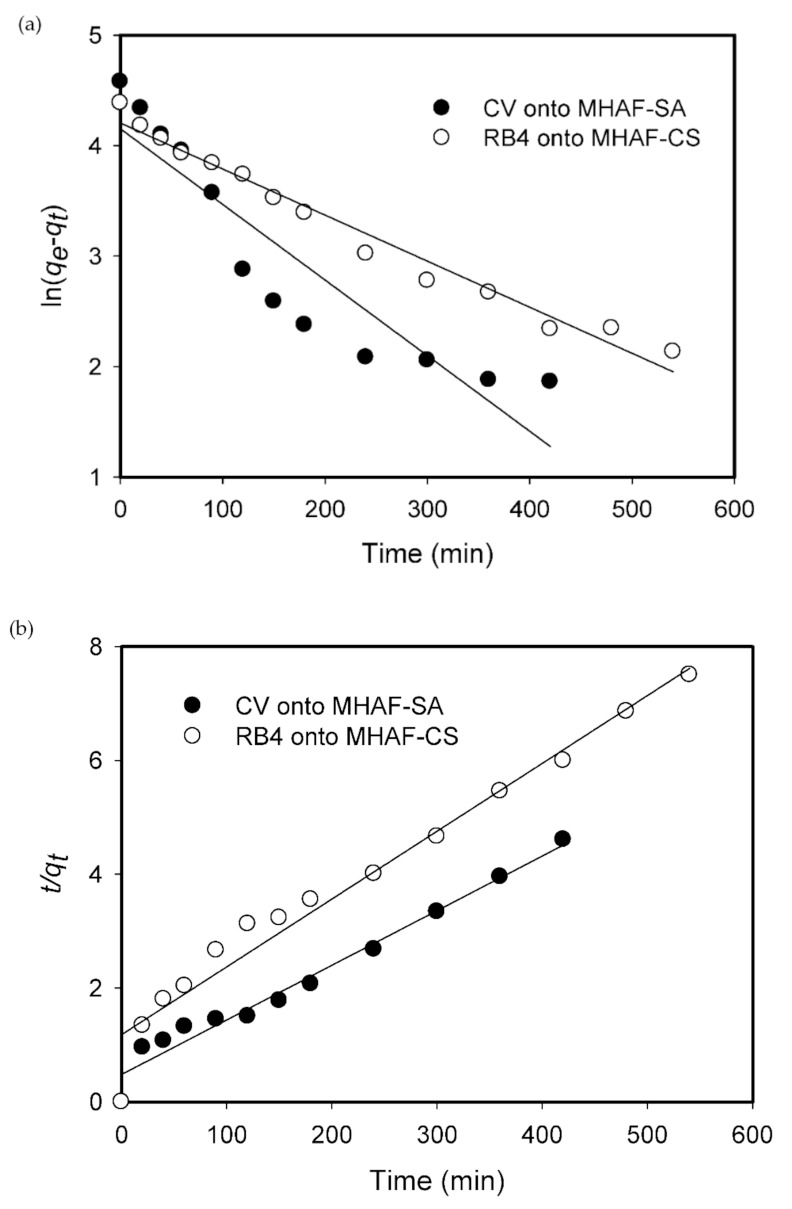
Kinetics models for dyes adsorption: (**a**) pseudo-first-order model, (**b**) pseudo-second-order model.

**Figure 9 molecules-26-01437-f009:**
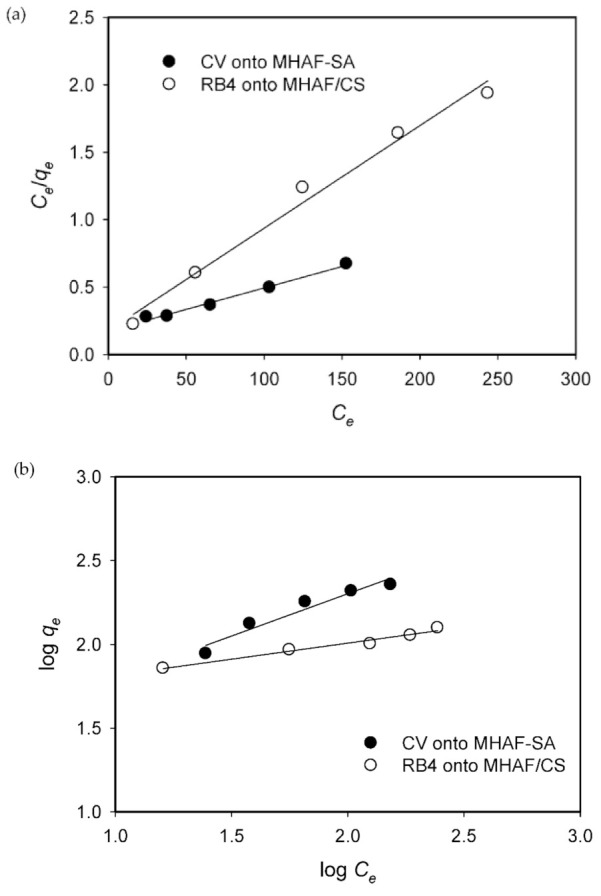
Adsorption isotherm models for dyes adsorption at 40 °C: (**a**) Langmuir, (**b**) Freundlich.

**Figure 10 molecules-26-01437-f010:**
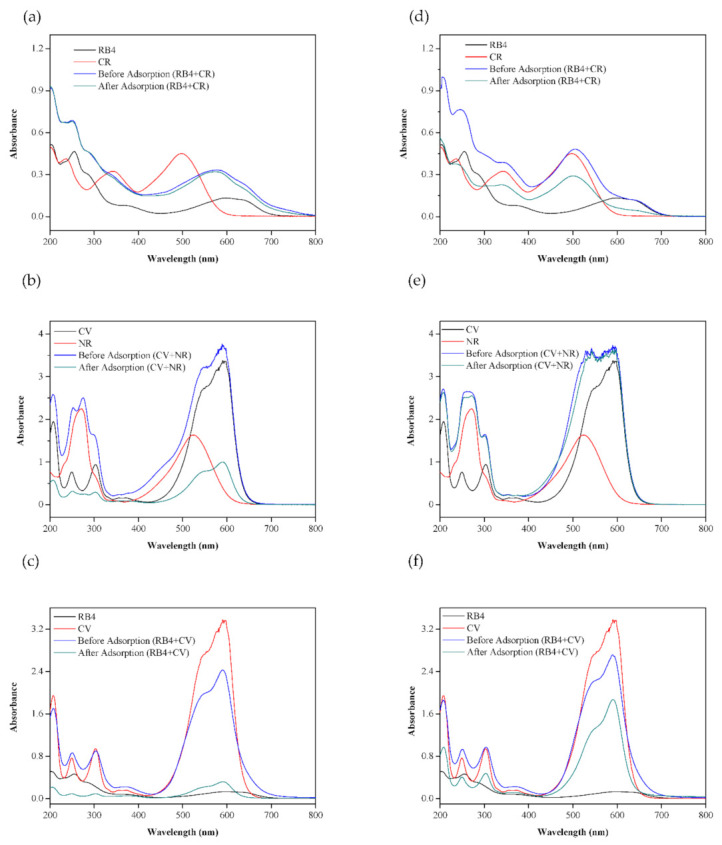
The absorption spectra of the dye mixture onto MHAF-SA (**a**–**c**) and MHAF-CS (**d**–**f**).

**Figure 11 molecules-26-01437-f011:**
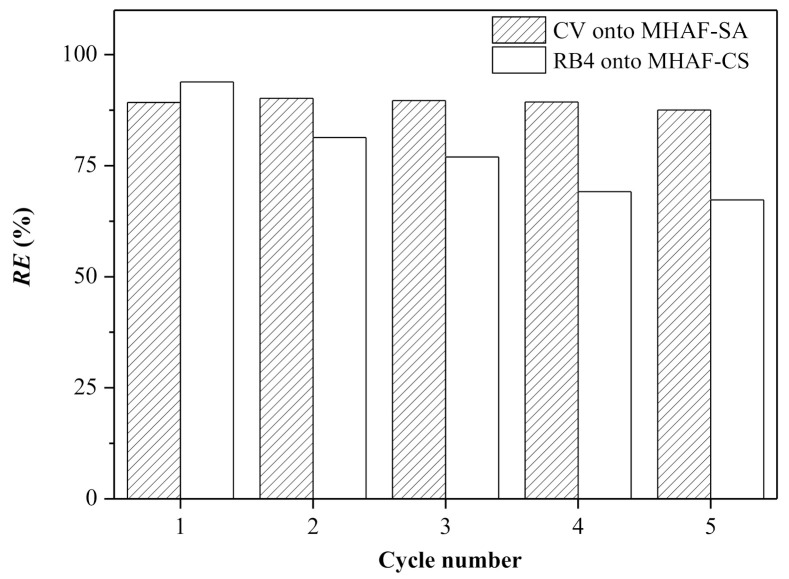
Regeneration performance of MHAF-SA and MHAF-CS.

**Figure 12 molecules-26-01437-f012:**
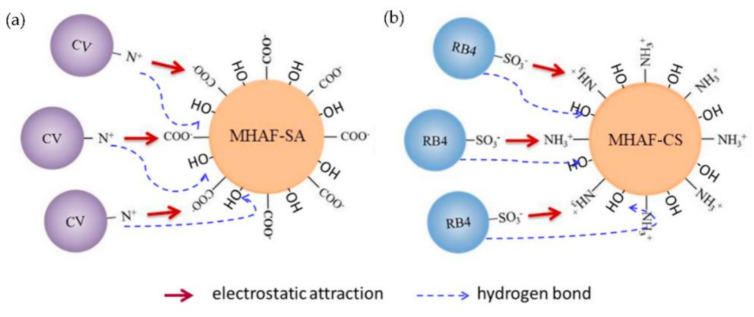
Adsorption mechanism of (**a**) CV onto MHAF-SA and (**b**) RB4 onto MHAF-CS.

**Figure 13 molecules-26-01437-f013:**
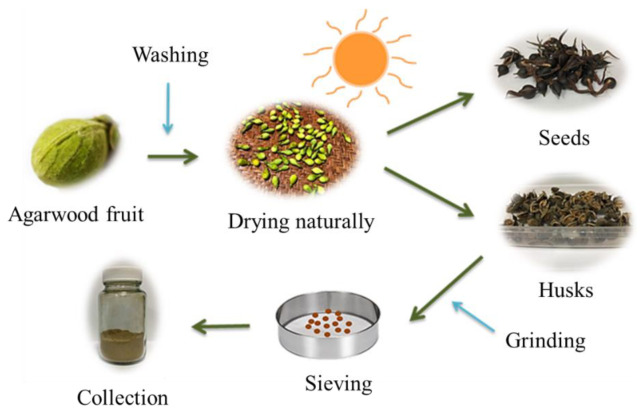
The treatment process of agarwood fruit.

**Table 1 molecules-26-01437-t001:** Kinetic parameters for the adsorption of dyes onto hydrogel beads at 40 °C.

Dye	CV	RB4
*q_exp_* (mg/g)	97.56	80.41
**PFOM**		
*k_1_* (1/min)	0.0068	0.0042
*q_e_* (mg/g)	63.7	83.0
*R* ^2^	0.8353	0.9739
**PSOM**		
*k*_2_ (g/mg·g)	1.92 × 10^−4^	1.20 × 10^−4^
*q_e_* (mg/g)	104.17	84.03
*R* ^2^	0.9715	0.9635

**Table 2 molecules-26-01437-t002:** Isotherm parameters for dye adsorption onto hydrogel beads.

Dye	CV			RB4		
**Temperature (°C)**	**30**	**40**	**50**	**30**	**40**	**50**
**Langmuir**						
*q_m_* (mg/g)	232.56	312.50	370.37	156.25	222.22	270.27
*K_L_* (L/mg)	0.022	0.018	0.016	0.027	0.014	0.013
*R* ^2^	0.975	0.989	0.982	0.998	0.995	0.998
**Freundlich**						
*K_F_* (mg/g)(L/mg)^1/n^	22.43	19.60	16.39	37.13	42.09	56.87
*n*	2.41	1.98	1.74	6.84	5.21	4.90
*R* ^2^	0.875	0.936	0.957	0.983	0.974	0.858

**Table 3 molecules-26-01437-t003:** Comparison of adsorption capacity of various adsorbents.

Dye	Adsorbent	Adsorption Capacity (mg/g)	Reusability (Five Cycle)	Reference
CV	NaOH-modified rice husk	44.87	-	[40]
	Chitosan magnetic composite microspheres	86.6	−1.6%	[41]
	Acid-activated sintering process red mud	336.3	-	[42]
	Chitosan pyrrole	150.16	-	[43]
	Cellulose-based adsorbent	182.15	>−25%	[44]
	MHAF-SA	232.56–370.37	−1.69%	This study
RB4	Heat-treated fungal biomass	156.9		[45]
	Bokbunja seed wastes	26.1		[46]
	rice bran/ Fe_3_O_4_	185.19	−4.6%	[9]
	Activated carbon	131.9		[47]
	rice bran/SnO_2_/Fe_3_O_4_	218.82	−12%	[48]
	MHAF-CS	156.25–270.27	~−20%	This study

## Data Availability

Data is contained within the article.

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
