# Peer review of "Husk of Agarwood Fruit-Based Hydrogel Beads for Adsorption of Cationic and Anionic Dyes in Aqueous Solutions"

_molecules, 2021, doi:10.3390/molecules26051437_

Round 1
Reviewer 1 Report
The authors fabricated hydrogel-based beads consisted of agarwood fruit husk, sodium alginate (SA), and chitosan (CS) for removal of crystal violet and reactive blue4. The originality of your research is the manufacturing and characterization of agarwood husk into the SA and CS based adsorbent. I think it would be possible to publish in a "molecules" journal if only the results that prove the adsorption ability of Agarwood Husk are added.
Comments.
- In 3.1., MHAF-SA is modified with acetic acid (line 177) and MHAF-CS is modified with sodium hydroxied (line 187). What is the role of ammonia solution (line 106) in 2.4? (In 2.4., before immersion in ammonia solution, it is indicated as HAF-CS)
- In 3.1. “The swelling ratio of MHAF-CS decreased with the addition of HAF, which is less hydrophilic than chitosan. Although the swelling ratio of MHAF-CS is lower than that of chitosan, it was better for adsorption performance since the introduction of HAF into chitosan improved the swelling strength and stability (Lines 208-212)”, but earlier, “The higher swelling property can improve dye adsorption since more water molecules can penetrate into the beads (line 205-206)”, it was swelled by chitosan, so I thought that chitosan's influence on dye adsorption was even greater.
- In 3.4., it was said that the adsorption equilibrium of MHAF-SA and MHAF-CS reached at 360min and 420min (line 285), but it seems to be good to specify how you checked.
- it is necessary to indicate the equation for adsorption kinetic models (pseudo-first-order/pseudo-second-order models) in 3.5.
- you need to specify what it means to follow the pseudo-first-order and pseudo-second-order models in 3.5. And I think there should be a corresponding graph.
- In 3.6., I think the equation for adsorption isotherm models (Langmuir/Freundlich models) should be indicated. And I think there should be a corresponding graph.
- 9(d), it is shown as a graph for MHAF-CS, not a graph for MHAF-SA. It seems that the text term needs to be corrected (lines 346, 348, 350).
- 8) There is no standard deviation overall.
Reviewer 2 Report
In this manuscript, authors prepared two kind of hydrogel beads MHAF-SA and MHAF-CS to remove cationic and anionic dyes in aqueous solutions. The adsorption conditions and adsorption isotherm data was studied. And these heads were proved to have effectiveness, selectivity and reusability. It is a data detailed work, but I don’t think authors successfully elucidated the novelty of this work. I don’t think it can be considered for publication before answering the follow questions in a convincing way.
- Why did authors choose crystal violet and reactive blue 4 as model dyes in this work? Is there any special structure or significance of these two dyes?
- Since the adsorption behavior of MHAF-SA and MHAF-CS is based on the ion pair interaction, can we speculate that MHAF-SA can absorb nearly all cationic dyes and MHAF-CS can absorb nearly all anionic dyes? If so, I don’t think the description of “excellent adsorption selectivity” is suitable.
- Since “agarwood is the most expensive wood in the world” as authors described in line 56, what about the price and productivity of the husk of agarwood fruit? Is it a suitable material for adsorbent?
- As described in part 2.2, the treatment process of agarwood fruit is not easy. I wonder what is the yield of this process and if it can be treated in factory.
- Why did authors choose the husk of agarwood fruit to make hydrogel? Is there any advantages? In line 58-60, “the functional groups in the husk of agarwood fruit can make it a valuable adsorbent for removing dyes from aqueous solutions”. I didn’t find these “functional groups”. I didn’t think it is irreplaceable in this work from this manuscript.
- In figure 5, why the swelling ratio of MHAF-SA is more than that of MHAF-CS?
- The abstract should be more clear, especially the first sentence. It seems that only one kind of hydrogel beads prepared by three component (HAF, CS and SA) was developed for removal of CV and RB4 at the same time.
- What about the effectiveness and reusability of other adsorbents used in the removal of dyes?
Round 2
Reviewer 2 Report
I think this manuscript is suitable for publication.
Author Response
Thanks the suggestions of the reviewer.